# Push-Pull Structures Based on 2-Aryl/thienyl Substituted Quinazolin-4(3*H*)-ones and 4-Cyanoquinazolines

**DOI:** 10.3390/molecules27217156

**Published:** 2022-10-22

**Authors:** Tatyana N. Moshkina, Emiliya V. Nosova, Julia V. Permyakova, Galina N. Lipunova, Ekaterina F. Zhilina, Grigory A. Kim, Pavel A. Slepukhin, Valery N. Charushin

**Affiliations:** 1Department of Organic and Biomolecular Chemistry, Chemical Engineering Institute, Ural Federal University, Mira St. 19, 620002 Ekaterinburg, Russia; 2I. Ya. Postovsky Institute of Organic Synthesis, Ural Branch of the Russian Academy of Sciences, S. Kovalevskaya Str., 22, 620108 Ekaterinburg, Russia

**Keywords:** quinazolin-4(3*H*)-one, 4-cyanoquinazoline, 2-(biphenyl)quinazoline, 2-thienylquinazoline, π-linker, fluorescence, donor–acceptor systems

## Abstract

Design and synthesis of 2-(aryl/thiophen-2-yl)quinazolin-4(3*H*)-ones and 4-cyano-2-arylquinazolines with Et_2_N-, Ph_2_N- or carbazol-9-yl- electron donating fragment are described. The key photophysical properties of these compounds have been studied by UV/Vis absorption and fluorescence spectroscopy in solvents of different polarity (toluene and MeCN). 2-(Aryl/thiophen-2-yl)quinazolin-4(3*H*)-ones show fluorescence in blue-green region in toluene solution with quantum yields up to 89% in the case of 2-(4’-N,N-diphenylamino[1,1’-biphenyl]-4-yl)-quinazolin-4(3*H*)-one. Moreover, triphenylamino derivative based on quinazolin-4(3*H*)-one with *para*-phenylene linker displays the highest quantum yield of 40% in powder. The fluorescence QY of Et_2_N and Ph_2_N derivatives decrease when going from toluene to MeCN solution, whereas carbazol-9-yl counterparts demonstrate strengthening of intensity that emphasizes the strong influence of donor fragment nature on photophysical properties. 4-Cyanoquinazolines are less emissive in both solvents, as well as, in solid state. The introduction of cyano group into position 4 leads to orange/red colored powder and dual emission bands. Some molecules demonstrate the increase in emission intensity upon addition of water to MeCN solution. According to frontier molecular orbitals (HOMO, LUMO) calculations, the energy gap of 4-cyanoquinazoline decreases by more than 1 eV compared to quinazolin-4-one, that is consistent with experimental data.

## 1. Introduction

Diazines and their benzoannulated counterparts represent an important class of chemical compounds applied in numerous areas of chemistry, medicine and technology. Over the last two decades, there has been a remarkable increase in the number of fluorescent diaza-heterocycles [1,2,3,4]. Among them quinoxaline and quinazoline derivatives are considered as pH sensors [5,6,7], luminescent detectors [8,9,10], imaging agents [11], components for organic light-emitting diodes (OLED) materials [12,13], solar cells and organic photovoltaic [14,15], and so on. The fundamental work has been done for the establishment of detailed structure–property relationships (SPRs) providing beneficial information for fine-tuning of the key properties and the rational design and synthesis of fluorophores. For example, the *meta*-linked D–A architecture in *m*TPA–phenanthroimidazole was applied for synthesis of near ultraviolet emissive materials [16]. The presence of rotatable groups is known to be favorable for aggregation induced/enhanced emission, and some of substituted benzodiazine derivatives display these properties [17,18]. The introduction of dibenzoannulated azines in the core, as well as attaching of cyano group to the ring, were demonstrated as effective approaches to develop thermally activated delayed fluorescent (TADF) emitters [19,20,21,22].

Quinazolin-4(3*H*)-one represents diaza-heterocycle with electronic structure similar to quinazoline core and can be used as an effective electron withdrawing fragment to design push-pull structures. Meanwhile, quinazolinone-based molecules of donor-acceptor type have not been properly explored, and the data is limited. However, carbazolyl-substituted quinazolinones **A** were established to be suitable components for fabrication host materials for green and blue phosphorescent organic light-emitting diodes (Figure 1) [23]. Another scientific group showed the ability of quinazolinone-based dyes **B** to act as fluorescent probes in human cells [24]. 2-Quinolinylquinazolin-4(3*H*)-one **C** demonstrated promising sensory properties towards zinc cations [25]; BF_2_ complexes of 2-pyridinyl quinazolin-4(3*H*)-ones **D** displayed strong luminescence both in solution and solid state, as well as large Stokes shift [26,27]. Moreover, 2-(2-hydroxyphenyl)quinazolines **E** are well known molecules of excited-state intramolecular proton-transfer (ESIPT) type molecules for imaging and detecting purposes [28]. 

Our research group is working on different 2,4-disubstituted quinazoline fluorophores and investigating the structure-property relations and possible practical applications. Previously, the π-conjugated chromophores with 4-(morpholin-4-yl)quinazoline electron withdrawing part have been obtained, and the influence of nature and length of π-linker on photophysical properties have been studied [29,30]. We succeeded in deepening of our understanding of the pH- and solvent-dependent behavior of these compounds. Moreover, the controlled protonation for generation of white light emission has been demonstrated [30]. Incorporation of 4-cyanoquinazoline or quinazolin-4(3*H*)-one cores allows to reinforce the electron-withdrawing properties. We described synthesis and photophysical properties of 4-cyano-2-thienylquinazoline derivatives [31], and we confirmed the strong electron withdrawing nature of 4-cyanoquinazoline core. On the other hand, quinazolin-4(3*H*)-one bearing 5-(4-diethylaminophenyl)thiophen-2-yl substituent at the position 2 possesses strong photoluminescence in the solution [32].

Herein we expend the series of quinazolin-4(3*H*)-ones and their 4-cyano counterparts by the synthesis of chromophores with different π-linkers and arrangement of substituents to handle detailed investigation of SPRs and to discover preferable structures for further practical applications. In the frames of this work, we study key photophysical properties in two solvents (toluene and MeCN). Additionally, we analyze emission behavior in solid state and in MeCN/water mixtures and calculate HOMO/LUMO energy levels. 

## 2. Results

### 2.1. Synthesis

The synthesis of quinazolinones **4**–**6** is summarized in Figure 1. Fluorophores **4**, **5** and **6** have been obtained by Suzuki-Miyaura cross-coupling of corresponding 2-bromothienylquinazolin-4(3*H*)-one **1**, 4(3)-bromophenylquinazolin-4(3*H*)-ones **2**, **3** with arylboronic acid or arylboronic acid pinacol ester under typical conditions in moderate to good yields (49–84%), Figure 1. The starting bromine derivatives **1**–**3** for cross-coupling reactions were obtained and described previously [30,32]. 

Synthetic approach to 4-cyano-2-thienylquinazolines **7** was developed by our research group earlier [31,32]. It includes nucleophilic substitution of chorine atom at position 4 of quinazoline core by the CN group in DMF under heating and subsequent Suzuki-Miyaura cross-coupling with arylboronic acid. The same route was used for novel π-conjugated chromophores **8** and **9** with 1,4-phenylene or 1,3-phenylene linker (Figure 2) starting from described 2-(4(3)-bromophenyl)-4-chloroquinazolines (yields 13–56%) [30]. Remarkably, cyanation of 4-chloroquinazolines with potassium cyanide results in 4-cyano intermediates in 92% and 94% yield, and the yields are higher compared to 2-thienyl counterpart [32]. 

Finally, we have synthesized 4-cyano-2-(4-diphenylaminophenyl)quinazoline **11** with shortened π-system compared to derivative **8b** (Figure 2). Quinazolin-4(3*H*)-one **10** underwent chlorination and subsequent nucleophilic substitution by cyanide residue. We failed to isolate the intermediate 4-chloroquinazoline. Moreover, the yield of target product **11** was 28%. Probably the strong electron-donating effect of the TPA unit causes the chlorination step to be sluggish, leading to the low yield of **11**. 

The target chromophores **4**–**6**, **8**, **9** as well as **11** were characterized by ^1^H, ^13^C NMR spectroscopy, mass spectrometry and elemental analysis data. ^13^C NMR data for sample **6a** has not been obtained due to its poor solubility in organic solvents (including DMSO-d_6_ under heating). 

Single crystals of quinazoline **11** were obtained by a slow evaporation technique (*n*-hexane/chloroform mixture as a solvent) and analyzed by X-ray diffraction analysis (XRD). According to XRD data, the compound is crystallized in the centrosymmetric space group of the triclinic system (Figure 3a).

The mean bond distances and angles of the molecule (Appendix A) are near to standard values. The molecule is non-planar, and Ph-substituents of the NPh_2_-groups are turned on the high angle toward the plane of the nitrogen atom (Figure 2b and Appendix A). The nitrogen of the C_6_H_4_-NPh_2_ group is practically planar. The atom N is deviated from the plane C(13)C(23)C(14) on the 0.054 Å and demonstrates significant conjugation with phenylene moiety (distances N(4)C(14) = 1.402(3), N(4)C(13) = 1.433(3), N(4)C(23) = 1.430(2) Å). Any significantly shortened intermolecular contacts or any special packing in the crystal were not observed.

### 2.2. UV/Vis and Fluorescence Spectroscopy

UV/Vis and photoluminescence (PL) spectra of compounds **4b**,**c**, **5a,b**, **6a,b**, **8a,b** and **9**–**11** in toluene (ε_r_ = 2.38) [33] and acetonitrile (ε_r_ = 36.64) [33] are presented in Appendix A. The main characteristics for described **4a**, **7a–c** and novel fluorophores are combined in Table 1. 

In general, the positions of the longest wavelength absorption bands of quinazolin-4(3*H*)-ones **4**–**6**, **10** are observed in the UV or purple region (292–410 nm) depending on donor and π-linker nature, arrangement of substituents at phenylene ring as well as solvent polarity. Considering the electron donating influence we note that Et_2_N- and Ph_2_N- counterparts in series **4** and **5** display similar position of absorption maxima (λ_abs_ = 410 nm for **4a**, λ_abs_ = 405 nm for **4b**, λ_abs_ = 370 nm for **5a** and λ_abs_ = 370 nm for **5b** in toluene (Table 1, Figure 4a). Carbazolyl-derivatives **4c** and **5c** show typical considerable hypochromic shift with maxima at 370 nm (**4c**) and 342 nm (**5c**) compared to their diethyl and diphenyl counterparts (Table 1, Figure 4a), that can be explained by shortened D–A conjugation due to rigid structure of carbazolyl unit and twisted configuration of molecule. The results are consistent with our previous research on quinazolines of linear structure [30] and with literature data [34,35]. The change of solvent polarity (from toluene to MeCN) leads to a hypsochromic shift of absorption maximum by 3–10 nm (compounds **4a–c**, **5a–c**). 

The influence of π-linker nature on absorption band is shown on Figure 4b. 2-Thienylquinazolin-4(3*H*)-one **4b** displays strong absorption band around 410 nm in toluene which is bathochromically shifted relative to other diphenylaminoquinazolines **5b, 6b** and **10**. Compounds **5b** and **10** display similar shape and position of absorption band both in toluene and MeCN solution (Figure 4b, Table 1). 2-(*Meta*-triphenyl)phenylquinazolin-4(3*H*)-one **6b** shows broad absorption band around 328 nm which is hypsochromically shifted by 42 nm in toluene and 32 nm in MeCN compared to para-substituted 2-phenylquinazoline **5b** (Table 1). Probably, a meta-linked D–A architecture results in breaking the conjugation between D and A and reflects in blue-shifted absorption. Worse conjugation in bipolar molecules containing meta-phenylene linker and larger dihedral angle, leading to increased steric hindrance, as well as shortened electron delocalization, were reported [36].

Studied quinazolin-4(3*H*)-ones **4**–**6**, **10** exhibit fluorescence maxima in the range from 430 nm to 490 nm in toluene and from 408 to 560 nm in MeCN. The influence of solvent polarity on emission is considerable, that emphasizes the high polarized excited state of the chromophores. As we expected, 2-thienylquinazolin-4(3*H*)-one with Et_2_N group **4a** possessed the most longwave emission in toluene. When going to Ph_2_N and then to carbazolyl counterparts (**4b** and **4c**, respectively) we observed shift to shorter wavelength, and the introduction of the carbazolyl unit has a more significant impact. The same correlation was noted for the second series, 2-(4-aryl)phenylquinazolinones **5a–c** (Figure 4a) and, more or less, for 2-(3-aryl)phenylquinazolinjnes **6a–c**. The replacement of thienylene linker with phenylene one also leads to a hypochromic shift of emission (for example, chromophores **5b**, **6b** and **10** in comparison with **4b**, Figure 4b). 

The quantum yields of chromophores measured by relative method vary over a wide range: for example, QY of thienylene-contained chromophores reaches 82% in toluene (in case of compound **4b**). 2-Phenyl-derivatives **5a** and **5b** demonstrate the increase in QY in both solvents compared to counterparts **4a** and **4b** with the strongest emission for chromophore **5b** (Φ_F_ = 89%). The *meta*-phenylene quinazolin-4(3*H*)-ones **6a–c** is less emissive with the QY up to 23% (compound **6b** in toluene). Notably, the strongest emission is observed for toluene solution of diphenyl-containing quinazolinones **4b**, **5b** and **6b** in each series. When going to MeCN, the QY decreases in the case of Et_2_N and Ph_2_N derivatives, probably, due to stabilization of intramolecular change transfer state and enhancement of non-radiative decay. Contrary, carbazolyl-derivatives **4c**, **5c** and **6c** demonstrate opposite result. Such an influence of the carbazolyl-phenyl residue may be due to the destruction of the conjugation resulting from the withdrawal of the phenylene fragment from the quinazoline plane as it was previously noted for 4-(4-(9*H*-carbazol-9-yl)phenyl)-2-phenyl and 2-quinolyl quinazolines [37,38].

The introduction of the CN group at position 4 of the quinazoline core led to great change in photophysical properties. In absorption spectra of **7a–c** and **8a,b,** we observed the main longwave band at 365–402 nm and weak band as a shoulder at 415–475 nm in toluene (Figure 5a). Upon excitation the chromophores display emission band with two peaks (Table 1, Figure 5a). The presence of shoulder-type peak in absorption spectra and double peak emission may be raised from the formation of dimers or aggregation between chromophore and solvent molecules. The absorption band of *meta*-phenylene derivative **9** is hypsochromically shifted in comparison with thienylene counterpart **7b** due to less effective conjugation of bi-phenyl moiety and large twisting of the phenylene ring with neighboring units in the π-bridge. The intense peaks of 4-cyanoquinazoline **11** at 367 nm in toluene and 360 nm in MeCN originate from transitions of the main conjugated skeleton, and the weaker peak at 432 and 415 nm is attributed to charge transfer (CT) transitions (Figure 5). The fluorescence quantum yields of CN-derivatives are lower compared to quinazolinone counterparts and do not exceed 23% in toluene. In more polar solvent (MeCN) the values of QY decrease; in the case of compound **11** the quenching of luminescent properties is observed. 

In comparison with 4-(morpholin-4-yl)quinazolines [30] quinazolin-4(3*H*)-one derivatives show bathochromic shift in both absorption and emission maxima that ascribes to better conjugation and stronger electron accepting quinazolinone fragment. 

The time-resolved emission data were measured to further characterize the emission bands. The average lifetimes are presented in Table 2 and more detailed measurements are combined in Appendix A. The lifetimes were mono-, bi- or three-component depending on structure and solvent polarity. In toluene, the compounds **4a,b** demonstrate the close values of 1.94 ns and 1.90 ns, respectively. The emission lifetime of carbazolyl derivative **4c** is lower (0.75 ns). Quinazolines **5a–c** with 1,4-phenylene moiety show the similar correlation between electron donating part nature and emission lifetime. In MeCN solution, the lifetime of both series **4** and **5** increases, reaching 2.54–3.66 ns (Table 2). *Meta*-substituted compounds **6b,c** and 2-diphenylaminophenylquinazolin-4(3*H*)-one counterpart **14** display longer luminescence lifetime in MeCN compared to the toluene solution. 

The lifetimes of 4-cyano derivatives are more complicated, probably, due to the formation of dimers, in addition to structural and solvent reasons. The scope of fluorescence lifetime data predicts the existence of several fluorescent species corresponding to locally exited (LE), intramolecular charge transfer (ICT), twisted intramolecular charge transfer (TICT) states [39] for some quinazoline derivatives that requires more carefully experiments in different media, concentrations and so on, and these investigations are in the progress. 

### 2.3. Photoluminescence Data for Compounds ***4***–***7*** and ***8***–***11*** in Solid State and in MeCN/Water Mixture

Synthesized fluorophores **4**–**11** exhibit emission in the solid state under UV-irradiation (Figure 6). 

Further, the emission spectra of these compounds were recorded, and the data are presented in Table 3 and Appendix A. The quinazolinone derivatives **4a–c**, **5a–c**, **6a–c** and **10** possess dark blue (**6c**) to yellow (**10**) emission with maxima from 407 to 537 nm (Figure 7а and Appendix A). The commission international de L’Eclairage (CIE) coordinates from (0.19, 0.13) to (0.14, 0.21) (Figure 7b) were observed for fluorophores **5а-с**, **6b,с** demonstrating blue emission. In general, when going from solution to solid state, we observed blue shift of emission maximum for derivatives **4a–c**, **5a–c**, **6a–c** to blue region (Table 3).

In terms of the chemical structure, the observed differences in solid-state emission appeared to depend on the electron donating residue and π-linker nature, and substituents arrangement (para or meta). Diphenylamino-derivatives **4b**, **5b** and **6b** demonstrate the strongest emission intensity in their series **4**, **5** and **6**. Linear non-planar chromophores **5a–c** with 1,4-phenylene spacer show the highest quantum yield compared to thienylene and 1,3-phenylene-containing counterparts **4a–c** and **6a–c**, respectively. The differences can be ascribed to the changing dihedral angle between fragments in the molecule, intermolecular interactions, and intermolecular packing modes caused by the variation in their terminal group and π-linker. Probably, phenylene moiety is turned from the plane of molecule, and the twisted structure prevents close packing that leads to blue shifted emission and intensity enhancement regarding to thienylene counterparts.

The introduction of the CN-group into position 4 of quinazoline core leads to red/orange-colored compounds with poor emission under UV-light (Figure 6). The emission spectra of cyano-derivatives **7a–c** [31], **8a,b** and **9** display two maxima with low intensity (Table 2). The dual emission can be ascribed to stabilization of two lowest exited states (ICT and TICT) caused by dimer formation through the CN group [32]. The quinazoline chromophore **11** with shortened π-conjugation demonstrates only one emission peak in powder and the highest QY of 3%. According to X-ray analysis, any significantly shortened intermolecular contacts have not been observed and, probably, the molecule forms the only exited state. Notably, the threefold increase in QY value was observed on passing from quinazolinone **10** to its cyano-counterpart **11** (Table 3). 

It is well known that molecules displaying aggregation induced emission (AIE) have enormous potential for practical application. Frequently, AIE-active compounds show mechanochromic properties [40]. Generally, AIE phenomena can be caused by restriction of intramolecular motions (RIM) or TICT-state inhibition accompanied by aggregation [41]. By manipulating the aggregation/disaggregation process, various fluorescence turn-on probes based on AIE-active luminophores have been fabricated for the sensitive detection of different analytes.

To study the AIE phenomenon for chromophores, we have analyzed emission behavior of some samples upon addition of water to MeCN solution. For the experiment, we chose Ph_2_N-containing chromophores **5b** and **6b** that demonstrated the highest emission in solid state. Quinazoline **11,** being not emissive in MeCN and possessing luminescence properties in solid, was analyzed, as well. Compounds **5b** and **6b** emit in pure MeCN at 535 and 560 nm, respectively. The addition of the first portion of water (up to 60%) to solution of **5b** leads to a gradual decrease in intensity and red shift of maximum caused by the increase in solvent polarity, and probably to stabilization of non-radiative TICT state (Appendix A). At 65% water content, the blue shifted maximum appears at 470 nm and the intensity increase with further addition of water. More intensive emission was observed at 80% water fraction compared to pure MeCN, and further slight decrease up to initial intensity was noticed. The compound **6b** possesses poor emission in pure MeCN. The luminescence is fully quenched after the addition of 10% of water and appears at 70% with blue-shifted maximum (Appendix A). At high water content, the plot of I/I_0_ versus water fraction has a similar profile to that of quinazoline **5b**. Compound **11**, which is non-emissive in pure MeCN, demonstrates gradual emission enhancement after the addition of 75% of water (Appendix A). The appearance of a blue-shifted emission band at a large amount of water in the case of derivatives **5b** and **6b** is probably caused by nanoaggregates formation. Moreover, the emission was even stronger relative to the original MeCN solution when the water content is larger than 90%. To clarify the difference in behavior of triphenylamino quinazoline derivatives **5b**, **6b**, **11** in MeCN/water mixtures, a more detailed study of aggregation processes is required, which is beyond the scope of this research.

### 2.4. Quantum-Chemical Calculations

Further, we performed the DFT calculations of quinazolin-4(3*H*)-ones **4a–c**, **5a–c**, **6a–c**, **10** and 4-cyanoquinazolines **8a,b**, **9**, **11** in gas phase at the B3LYP/6–311 G* level using the Orca 4.0.1 software package [42,43,44,45,46] and conducted the chemical optimization on their energy levels based on DFT/B3LYP/6-31G (d,p) using Gaussian 09. The HOMO and LUMO energy levels of compounds **4**–**6**, **10** are in the range of −5.20 to −5.80 eV and −1.65 to −2.25 eV, respectively, with an energy band gap (ΔE) ranging from 3.20 to 4.05 eV (Figure 8). The smallest band gap is observed for dye **4b** bearing 5-(4-diphenylaminophenyl)thiophenylene substituent. Introduction of the CN group in the position 4 of quinazoline core leads to a significant decrease in the band gap (2.33–2.61 eV), mainly due to lowering of LUMO energy (Figure 8). The energy levels of quinazolinone derivatives **4**–**6**, **10** have changed slightly regarding to 4-morpholinyl counterparts [30].

The spatial distributions of the calculated HOMO and LUMO energy levels of fluorophores are shown in Appendix A. The HOMO of 2-aryl/thienylquinazolin-4(3*H*)-ones **4a–c**, **5a–c**, **6a–c**, **10** is basically distributed over the electron donating diethylaminophenyl, diphenylaminophenyl or (carbazol-9-yl)phenyl unit, and the LUMO of 2-aryl/thienyl quinazolin-4(3*H*)-ones is mainly localized at quinazolin-4(3*H*)-one unit and its neighboring 2-phenyl or 2-thienyl substituent. In the case of 4-cyano derivatives, the localization of HOMO level is similar to 4-oxo counterparts, whereas LUMO is distributed on 4-cyanoquinazoline fragment (Appendix A), probably the strong electron deficient cyano substituent stabilizes the excited state. When compared to the quinazolinones derivatives, the HOMO and LUMO distributions of 4-cyanoquinazolins are barely overlapped.

## 3. Experimental Methods

### 3.1. General Information

Unless otherwise indicated, all common reagents and solvents were used by commercial suppliers without further purification. Melting points were determined on Boetius combined heating stages. ^1^H NMR and ^13^C NMR spectra were recorded at room temperature at 400 and 100 MHz respectively, on a Bruker DRX-400 spectrometer; or at 151 MHz on a Bruker DRX-600 spectrometer (^13^С NMR spectra for compounds **4b,c**). Hydrogen chemical shifts were referenced to the hydrogen resonance of the corresponding solvent (DMSO-d_6_, δ = 2.50 ppm or CDCl_3_, δ = 7.26 ppm). Carbon chemical shifts were referenced to the carbon resonances of the solvent (DMSO-d_6_, δ = 39.5 ppm CDCl_3_, δ = 77.2 ppm). Peaks are labeled as singlet (s), doublet (d), triplet (t), quartet (q) and multiplet (m). Mass spectra were recorded on the SHIMADZU GCMS-QP2010 Ultra instrument with electron ionization (EI) of the sample. Microanalyses (C, H, N) were performed using the Perkin–Elmer 2400 elemental analyzer. 

### 3.2. Photophysical Characterization

UV/Vis spectra were recorded with Shimadzu UV-2600 spectrophotometer. Photoluminescent spectra were recorded on a Varian Cary Eclipse spectrofluorometer. UV/Vis and fluorescence spectra of solutions were recorded using standard 1 cm quartz cells at room temperature. The Ф_F_ values were calculated using the established procedure with 3-aminophalimide in ethanol (Ф_F_ = 0.60) and in quinine sulfate 0.1N H_2_SO_4_ (Ф_F_ = 0.55) [47]. Fluorescence spectra in a solid state were measured by the integrating sphere Quanta-φ F-3029 at Horiba FluoroMax-4.

The emission lifetimes have been measured using the TCSPC option of FS5 Edinburgh Instruments spectrofluorometer. The sample has been excited by EPLED-300 picosecond pulsed light emitted diode centered at 300 nm and EPL-375 picosecond pulsed diode laser centered at 375 nm. The instrument response function (IRF) has been recorded under described conditions by replacing the sample with a silica diffuser. The time decay data have been analyzed by nonlinear least-squares fitting with the deconvolution of the IRF using the Fluoracle software package.

Analytical experiments were carried out by using equipment from the Center for Joint Use «Spectroscopy and Analysis of Organic Compounds» at the Postovsky Institute of Organic Synthesis of the Ural Branch of the Russian Academy of Sciences.

### 3.3. Crystallography

The single crystal (red block of 0.45 × 0.35 × 0.25) of compound **11** was used for X-ray analysis. The single crystals XRD experiment was performed for compound **11** on an automated diffractometer “Xcalibur 3” with a standard procedure (graphite-monochromated Mo Kα-irradiation, T = 295(2) K, ω-scanning with step 1°). An empirical absorption correction was applied. Using Olex2 [48], the structure was solved with the ShelXT structure solution program using Intrinsic Phasing and refined with the ShelXL [49] refinement package using full-matrix Least Squares minimization. All non-hydrogen atoms were refined in an anisotropic approximation; the H-atoms at the C(5)–C(8) carbons were solved by direct method and were refined independently in the isotropic approximation, all other H-atoms were placed in the calculated positions and refined isotropically in the “rider” model.

Crystal data for **11** C_27_H_18_N_4_, M = 398.45, triclinic, a = 9.1982(9) Å, b = 9.8103(10) Å, c = 12.9224(11) Å, α = 96.908(8)°, β = 107.028(8)°, γ = 105.834(9)°, V = 1046.95(18) Å^3^, space group P–1, Z = 2, μ(Mo Kα) = 0.076 mm^−1^. On the angles 3.61 < 2Θ < 30.98°, 8589 reflections measured, 5685 unique (R_int_ = 0.0528) which were used in all calculations. Goodness to fit at F^2^ 0.991; The final R_1_ = 0.1425, wR_2_ = 0.2096 (all data) and R_1_ = 0.0668, wR_2_ = 0.1481 (I > 2s(I)). Largest diff. peak and hole 0.263 and −0.203 ēÅ^−3^.

The result of X-ray diffraction analysis for compound **11** was deposited in the Cambridge Crystallographic Data Centre (CCDC 2141460). The data is free and can be available at www.ccdc.cam.ac.uk (accessed on 12 January 2022).

### 3.4. Preparation of Intermediates

Starting quinazolinones **1**, **2**, **3** and **10,** as well as the corresponding 4-chloro derivatives, were synthesized as described in our previous works [30,32]. 

2-(4-Bromophenyl)- and 2-(3-bromophenyl)-4-cyanoquinazolines (bromo derivatives **I** and **II**) were obtained similar to described procedure [32]. Freshly prepared potassium cyanide (0.25 g, 3.8 mmol) and sodium *p*-toluenesulfonate (0.20 g, 1.0 mmol) were added to a solution of 4-chloro derivative (3 mmol) in dry DMF (12 mL). The mixture was heated at 95 °С for 3 h. After cooling the precipitate was filtered off and washed with water (30 mL) and EtOH (5 mL).

2-(4-Bromophenyl)-4-cyanoquinazoline (I): pale brown solid, yield 92%; mp > 300 °C; ^1^H NMR (DMSO-d_6_, 400 MHz) δ 7.74 (2H, d, ^3^*J* = 8.3 Hz, H-3′, H-5′), 7.95 (1H, m), 8.20 (2H, m), 8.26 (1H, d, ^3^*J* = 8.3 Hz), 8.48 (2H, d, H-2′, H-6′, ^3^*J* = 8.3 Hz); EIMS *m*/*z* 312 [M + 3]^+^ (17), 311 [M + 2]^+^ (92), 310 [M + 1]^+^ (18), 309 [M]^+^ (100), 285 (10), 259 (13), 257 (14), 178 (20), 151 (15), 115 (27), 102 (45), 101 (13), 76 (43), 75 (32), 51 (14), 50 (30); anal. C 58.12, H 2.58, N 13.57%, calcd for C_15_H_8_BrN_3_ (310.15), C 58.09, H 2.60, N 13.55%.

2-(3-Bromophenyl)-4-cyanoquinazoline (II). Pale brown solid, yield 94%; mp > 300 °C; ^1^H NMR (DMSO-d_6_, 400 MHz) δ 7.55 (1H, m, H-5′), 7.77 (1H, m, H-4′), 7.97 (1H, m), 8.22–8.28 (3H, m), 8.53 (1H, d, ^3^*J* = 7.8 Hz, H-6′), 8.65 (1H, s, H-2′); EIMS *m*/*z* 312 [M+3]^+^ (18), 311 [M + 2]^+^ (94), 310 [M + 1]^+^ (19), 309 [M]^+^ (100), 230 (40), 178 (27), 177 (11), 151 (19), 115 (31), 102 (50), 101 (14), 76 (52), 75 (38), 74 (11), 51 (19), 50 (40); anal. C, 58.13; H, 2.63; N, 13.59%, calcd for C_15_H_8_BrN_3_ (310.15), C 58.09, H 2.60, N 13.55%.

### 3.5. General Procedures of Suzuki Cross-Coupling

To mixture of bromo derivative **1**, **2**, **3**, **I** or **II** (0.65 mmol) in toluene (10 mL) the corresponding boronic acid or boronic acid pinacol ester (0.70 mmol), PdCl_2_(PPh_3_)_2_ (46 mg, 65 μmol), PPh_3_ (34 mg, 130 μmol), saturated solution of K_2_CO_3_ (3.7 mL) and EtOH (3.7 mL) were added. The mixture was stirred at 85 °C for 7–20 h in argon atmosphere in round-bottom pressure flask. The reaction mixture was cooled. After cooling, the target product was filtered off and washed with water and hexane [29,30,31].

### 3.6. Derivatives of 2-Arylthienylquinazolin-4(3H)-ones ***4***

2-(5-(4-N,N-Diphenylaminophenyl)thiophen-2-yl)quinazolin-4(3*H*)-one (4b). The reaction mixture was heated for 9 h. Green yellow solid, yield 81%; mp > 300 °C; ^1^H NMR (DMSO-d_6_, 400 MHz) δ 7.02–7.10 (8H, m), 7.29 (4H, m), 7.37 (1H, d, *^3^J* = 3.2 Hz), 7.42 (1H, m), 7.59 (3H, m), 7.73 (1H, m, CH quinaz.), 8.10 (1H, d, *^3^J* = 7.7 Hz), 8.17 (1H, m), 12.51 (1H, s, NH); ^13^С NMR (DMSO-d_6_, 151 MHz), δ 120.7, 122.2, 123.6, 123.7, 124.6, 125.9, 126.1, 126.4, 126.7, 126.8, 129.6, 130.5, 134.5, 135.0, 146.6, 147.7, 148.6, 161.7. EIMS *m*/*z* 473 [M + 2]^+^ (11), 472 [M + 1]^+^ (36), 471 [M]^+^ (100), 235 (13); anal. C 76.45, H 4.47, N 8.94%, calcd for C_30_H_21_N_3_S (471.57) C 76.41, H 4.49, N 8.91%.

2-(5-(4-(9*H*-Carbazol-9-yl)phenyl)thiophen-2-yl)quinazolin-4(3*H*)-one (4c). The reaction mixture was heated for 10 h. Dark yellow solid, yield 74%; mp > 290 °C; ^1^H NMR (DMSO-d_6_, 400 MHz) δ 7.28 (2H, m, 2 CH carbaz.), 7.40–7.48 (5H, m), 7.64–7.77 (4H, m), 7.76 (1H, m, CH quinaz.), 8.02 (2H, d, *^3^J* = 8.0 Hz), 8.13–8.19 (3H, m), 8.27 (1H, m), 12.63 (1H, s, NH); ^13^С NMR (DMSO-d_6_, 151 MHz) δ 109.7, 120.2, 120.5, 120.8, 122.8, 125.4, 125.9, 126.2, 126.8, 127.2, 127.4, 128.1, 128.8, 130.6, 131.9, 13.6, 136.6, 137.0, 139.9, 147.4, 148.5, 161.7; EIMS *m*/*z* 471 [M + 2]^+^ (11), 470 [M + 1]^+^ (36), 469 [M]^+^ (100), 350 (13), 235 (17), 92 (16), 91 (18); anal. C 76.76, H 4.11, N 8.93%, calcd for C_30_H_21_N_3_OS (469.56), C 76.74, H 4.08, N 8.95%. 

### 3.7. Biphenylene-Containing Quinazolin-4(3H)-ones ***5***, ***6***

2-(4’-N,N-Diethylamino[1,1’-biphenyl]-4-yl)quinazolin-4(3*H*)-one (5a). The reaction mixture was heated for 20 h. Grey solid, yield 69%; mp 265–267 °C; ^1^H NMR (CDCl_3_, 400 MHz) δ 1.22 (6H, t, *^3^J* = 7.0 Hz, 2 CH_3_), 3.43 (4H, q, *^3^J* = 7.0 Hz, 2 CH_2_), 6.78 (2H, d, *^3^J* = 8.7 Hz, H-3″, H-5″), 7.49 (1H, m, CH quinaz.), 7.58 (2H, d, *^3^J* = 8.7 Hz, H-2″, H-6″), 7.74–7.84 (4H, m, H-3′, H-5′, 2 CH quinaz.), 8.14 (2H, d, *^3^J* = 8.2 Hz, H-2′, H-6′), 8.33 (1H, d, *^3^J* = 8.0 Hz, CH quinaz.), 10.26 (1H, s, NH); ^13^C NMR (DMSO-d_6_, 100 MHz) δ 12.5 (2CH_3_), 43.7 (CH2), 111.8, 121.3, 124.7, 125.4, 125.8, 126.8, 127.5, 128.3, 132.9, 138.2, 142.0, 147.2, 150.1. EIMS *m*/*z* 370 [M + 1]^+^ (17), 369 [M]^+^ (59), 355 (28), 354 [M-СН_3_]^+^ (100), 326 [M-HNCO]^+^ (13), 177 (12), 119 (29); anal. C 78.04, H 6.28, N 11.36%, calcd for C_24_H_23_N_3_O (369.47), C 78.02, H 6.27, N 11.37%.

2-(4’-N,N-Diphenylamino[1,1’-biphenyl]-4-yl)-quinazolin-4(3*H*)-one (5b). The reaction mixture was heated for 10 h. Pale yellow solid, yield 84%; mp 265–267 °C; ^1^H NMR (CDCl_3_, 400 MHz) δ 7.09 (2H, m), 7.18 (6H, m), 7.28–7.34 (4H, m), 7.52 (1H, m, CH quinaz.), 7.58 (2H, d, *^3^J* = 8.6 Hz, H-2″, H-6″), 7.79–7.87 (4H, m, 2 CH quinaz., H-3′, H-5′), 8.23 (2H, d, *^3^J* = 8.6 Hz, H-2′, H-6′), 8.36 (1H, d, *^3^J* = 8.0 Hz, CH quinaz.), 10.59 (1H, s, NH); ^13^C NMR (DMSO-d_6_, 100 MHz) δ 121.2, 122.9, 123.2, 124.1, 124.8, 125.5, 125.6, 126.7, 127.5, 128.2, 129.4, 132.9, 132.9, 133.6, 141.3, 146.8, 147.1, 149.6, 154.8, 164.9. EIMS *m*/*z* 466 [M + 1]^+^ (36), 465 [M]^+^ (100); anal. C 82.57, H 4.99, N 9.01%, calcd for C_32_H_23_N_3_O (465.56), C 82.56, H 4.98, N 9.03%.

2-(4’-9*H*-Carbazol-9-yl[1,1’-biphenyl]-4-yl)-quinazolin-4(3*H*)-one (5c). The reaction mixture was heated for 20 h. Grey solid, yield 75%; mp 287–289 °C; ^1^H NMR (DMSO-d_6_, 400 MHz) δ 7.27 (2H, m, 2 CH carbaz.), 7.42 (5H, m), 7.67–7.74 (4H, m), 7.89 (2H, d, *^3^J* = 8.6 Hz), 8.03 (2H, d, *^3^J* = 8.6 Hz), 8.13–8.19 (3H, m, 2 CH phenylene, CH quinaz.), 8.45 (2H, d, *^3^J* = 8.6 Hz), 10.59 (1H, s, NH); ^13^C NMR (DMSO-d_6_, 100 MHz) δ 109.73, 120.1, 120.5, 122.8, 126.1, 126.3, 127.1, 128.3, 128.5, 131.1, 136.2, 139.0, 139.6, 140.1, 151.6. MS *m*/*z* 464 [M + 1]^+^ (38), 463 [M]^+^ (100), 344 (21), 119 (17); anal. C 82.90, H 4.59, N 9.06%, calcd for C_32_H_21_N_3_O (463.54), C 82.92, H 4.57, N 9.07%. 

2-(4’-(Diethylamino)-[1,1’-biphenyl]-3-yl)quinazolin-4(3*H*)-one (6a). The reaction mixture was heated for 10 h. Additionally, the solid was washed with CH_2_Cl_2_. Dark grey solid, yield 49%; mp 245–247 °C; ^1^H NMR (DMSO-d_6_, 400 MHz) δ 1.19 (6H, t, *^3^J* = 6.9 Hz, 2 CH_3_), 3.42 (4H, d, *^3^J* = 6.9 Hz, 2 CH_2_), 6.73 (2H, d, *^3^J* = 7.6 Hz, H-3″, H-5″), 7.45–7.53 (2H, m), 7.61 (2H, d, *^3^J* = 7.6 Hz, H-2″, H-6″), 7.71–7.80 (3H, m), 8.08 (1H, d, *^3^J* = 7.9 Hz), 8.16 (1H, d, *^3^J* = 8.3 Hz), 8.39 (1H, s, H-2′), 12.53 (1H, s, NH); EIMS *m*/*z* 370 [M + 1]^+^ (15), 369 [M]^+^ (51), 344 (21), 355 (28), 354 (100), 326 (11), 177 (15), 119 (27); anal. C 78.00, H 6.25, N 11.36%, calcd for C_24_H_23_N_3_O (369.47), C 78.02, H 6.27, N 11.37%.

2-(4’-(Diphenylamino)-[1,1’-biphenyl]-3-yl)quinazolin-4(3*H*)-one (6b). The reaction mixture was heated for 10 h. Grey solid, yield 55%; mp 260–262 °C; ^1^H NMR (CDCl_3_, 400 MHz) δ 7.03 (2H, m), 7.14 (6H, m), 7.23–7.32 (5H, m), 7.60 (1H, m), 7.66 (2H, d, *^3^J* = 7.6 Hz, H-2″, H-6″), 7.71–7.79 (2H, m), 7.83 (1H, d, *^3^J* = 8.3 Hz), 8.18 (1H, d, *^3^J* = 7.6 Hz), 8.24 (1H, d, *^3^J* = 7.6 Hz), 8.46 (1H, s, H-2′), 11.81 (1H, s, NH); ^13^C NMR (DMSO-d_6_, 100 MHz) δ 121.1, 123.3, 124.5. 125.4, 125.8, 126.4, 126.5, 127.4, 128.0, 128.9, 129.2. 129.6, 133.4, 133.5, 134.5, 139.9, 147.0, 147.1, 148.8, 152.5, 162.5; EIMS *m*/*z* 466 [M + 1]^+^ (35), 465 [M]^+^ (100), 464 (12), 233 (12); anal. C 82.55, H 4.96, N 9.01%, calcd for C_32_H_23_N_3_O (465.56), C 82.56, H 4.98, N 9.03%.

2-(4’-(9*H*-Carbazol-9-yl)-[1,1’-biphenyl]-3-yl)quinazolin-4(3*H*)-one (6c). The reaction mixture was heated for 12 h. Grey solid, yield 81%; mp > 290 °C; ^1^H NMR (DMSO-d_6_, 400 MHz) δ 7.32 (2H, m, ^3^J = 6.8 Hz, 2CH carbaz.), 7.45–7.50 (4H, m), 7.55 (1H, m), 7.72 (1H, m), 7.78 (3H, m), 7.87 (1H, m), 8.03 (1H, d, *^3^J* = 7.6 Hz), 8.15–8.20 (3H, m), 8.27 (3H, m), 8.61 (1H, s, H-2′), 12.75 (1H, s, NH); ^13^C NMR (DMSO-d_6_, 100 MHz) δ 109.5, 119.9, 120.3, 121.0, 122.7, 125.7, 125.8, 126.1, 126.4, 126.9. 127.0, 128.5, 129.2, 129.4, 133.3, 134.3, 136.5, 138.4, 139.5, 140.0, 148.6, 152.0, 162.0; EIMS *m*/*z* 464 [M + 1]^+^ (37), 463 [M]^+^ (100), 344 (19), 232 (12), 119 (28), 92 (12); anal. C 82.90, H 4.55, N 9.05%, calcd for C_32_H_21_N_3_O (463.54), C 82.92, H 4.57, N 9.07, O 3.45%.

### 3.8. Biphenylene-Containing 4-Cyanoquinazolines ***8***, ***9***

2-(4’-(Diethylamino)-[1,1’-biphenyl]-4-yl)quinazoline-4-carbonitrile (8a). The reaction mixture was heated for 7 h. After cooling the EtOAc/water mixture (1:1, 10 mL) was added. The organic layer was separated, and the aqueous layer was extracted with additional EtOAc (2 × 10 mL). The organic extracts were combined, and the solvent was evaporated under reduced pressure. The product was purified by column chromatography (SiO_2_, hexane/EtOAc, 3/1). Red solid, yield 13%; mp 145–147 °C; ^1^H NMR (DCCl_3_, 400 MHz) δ 1.22 (6H, t, *^3^J* = 6.9 Hz, 2 CH_3_), 3.43 (4H, q, *^3^J* = 6.9 Hz, 2 CH_2_), 6.78 (2H, d, *^3^J* = 8.4 Hz, H-3″, H-5″), 7.61 (2H, d, *^3^J* = 8.4 Hz, H-2″, H-6″), 7.75 (3H, m, H-3′, H-5′, CH quinaz.), 8.02 (1H, m, CH quinaz.), 8.17 (1H, d, *^3^J* = 8.6 Hz, CH quinaz.), 8.24 (1H, d, *^3^J* = 7.8 Hz, CH quinaz.), 8.63 (2H, d, *^3^J* = 8.3 Hz, H-2′, H-6′); ^13^C NMR (CDCl_3_, 100 MHz) δ 12.8, 44.6, 112.0, 114.7, 123.0, 125.1, 126.3, 126.8, 128.2, 129.2, 129.3, 129.6, 133.8, 135.8, 143.6, 144.5, 147.9, 152.0, 161.3; EIMS *m*/*z* 379 [M + 1]^+^ (16), 378 [M]^+^ (56), 364 (29), 363 (100) [M-СН_3_]^+^, 335 (15), 334 (14), 306 (10), 181 (11); anal. C 79.29, H 5.38, N 14.77%, calcd for C_25_H_22_N_4_ (378.48), C 79.34, H 5.36, N 14.80%. 

2-(4’-(Diphenylamino)-[1,1’-biphenyl]-4-yl)quinazoline-4-carbonitrile (8b). The reaction mixture was heated for 7 h. The crude product was isolated from toluene solution similar to **8a**. The product was purified by column chromatography (gradually from hexane to hexane/EtOAc (1:19)) and recrystallization from CH_2_Cl_2_/hexane mixture. Red-orange solid, yield 56%; mp 105–107 °C; ^1^H NMR (DCCl_3_, 400 MHz) δ 7.06 (2H, m), 7.15–7.18 (6H, m), 7.29 (4H, m), 7.59 (2H, d, *^3^J* = 8.7 Hz, H-2″, H-6″), 7.77 (3H, m, H-3′, H-5′, CH quinaz.), 8.04 (1H, m, CH quinaz.), 8.19 (1H, d, *^3^J* = 8.7 Hz, CH quinaz.), 8.26 (1H, d, *^3^J* = 8.2 Hz, CH quinaz), 8.67 (2H, d, *^3^J* = 8.7 Hz, H-2′, H-6′,); ^13^C NMR (CDCl_3_, 100 MHz) δ 114.6, 123.1, 123.4, 123.7, 124.8, 125.1, 127.0, 128.0, 129.4, 129.5, 129.6, 133.9, 134.9, 135.9, 143.6, 143.8, 147.7, 148.0, 152.0, 161.0. EIMS *m*/*z* 475 [M + 1]^+^ (39), 474 [M]^+^ (100), 473 (10), 237 (13); anal. C 83.48, H 4.70, N 11.76%, calcd for C_33_H_22_N_4_ (474.18), C 83.52, H 4.67, N 11.81%.

2-(4’-(Diphenylamino)-[1,1’-biphenyl]-3-yl)quinazoline-4-carbonitrile (9). The reaction mixture was heated for 7 h. The crude product was isolated from toluene solution similar to **8a**. The product was purified by column chromatography (SiO_2_, gradually from hexane to hexane/EtOAc (1:19)) and recrystallization from CH_2_Cl_2_/hexane mixture. Red solid, yield 21%; mp 210–212 °C. ^1^H NMR (CDCl_3_, 400 MHz) δ 7.05 (2H, m), 7.16–7.21 (6H, m), 7.26-7.31 (4H, m), 7.61 (3H, m, H-2″, H-6″), 7.75–7.81 (2H, m), 8.05 (1H, m), 8.21 (1H, d, *^3^J* = 8.6 Hz), 8.27 (1H, d, *^3^J* = 8.3 Hz), 8.57 (1H, d, *^3^J* = 7.7 Hz), 8.85 (1H, s, H-2′). ^13^C NMR (CDCl_3_, 100 MHz) δ 114.6, 123.2, 123.3, 124.1, 124.6, 125.1, 127.2, 127.3, 128.1, 129.5, 129.6, 127.7, 129.9, 134.7, 135.9, 137.0, 141.5, 143.6, 147.7, 147.8, 151.9, 161.1; EIMS *m*/*z* 475 [M + 1]^+^ (38), 474 [M]^+^ (100); anal. C 83.49, H 4.72, N 11.78%, calcd for C_33_H_22_N_4_ (474.18), C 83.52, H 4.67, N 11.81%. 

### 3.9. 2-(4-(Diphenylamino)phenyl)quinazoline-4-carbonitrile (***11***)

To quinazolinone **10** (0.30 g, 0.77 mmol) POCl_3_ (1.0 mL, 10.72 mmol) was added. The mixture was refluxed for 2 h using drying tube containing anhydrous calcium chloride. After cooling, the mixture was poured into ice and formed precipitate was filtered off and washed with saturated NaHCO_3_ solution. To the crude 4-chloroquinazoline red solid in dried DMF (4.4 mL), KCN (0.1 g, 1.54 mmol) and sodium *p*-toluenesulfonate (0.10 g, 0.52 mmol) was added, the reaction mixture was refluxed at 95 °C for 3 h. After cooling the water (10 mL) was added and the precipitate was filtered off. The product was purified by column chromatography (SiO_2_, gradually from hexane/EtOAc (19:1) to EtOAc). Orange solid, yield 28%; mp 150–152 °C; ^1^H NMR (CDCl_3_, 400 MHz) δ 7.16–7.26 (8H, m), 7.38 (4H, m), 7.77 (1H, m, CH quinaz.), 8.04 (1H, m, CH quinaz.), 8.16 (1H, d, *^3^J* = 8.9 Hz, CH quinaz.), 8.26 (1H, d, *^3^J* = 8.9 Hz, CH quinaz.), 8.50 (2H, d, *^3^J* = 8.9 Hz, H-2′, H-6′); ^13^C NMR (CDCl_3_, 100 MHz) δ 114.7, 121.7, 122.8, 124.2, 125.1, 125.7, 128.9, 129.4, 129.6, 130.0, 135.7, 143.5, 147.1, 151.1, 152.0, 161.0; EIMS *m*/*z* 399 [M + 1]^+^ (31), 398 [M]^+^ (100), 397 (18); anal. C 81.35, H 4.58, N 14.01%, calcd for C_27_H_18_N_4_ (398.47), C 81.39, H 4.55, N 14.06%.

## 4. Conclusions

In summary, we have designed a series of push–pull chromophores bearing 2-(aryl/thiophen-2-yl)quinazolin-4(3H)-one and 4-cyano-2-arylquinazoline electron acceptor and Et_2_N-, Ph_2_N- or carbazol-9-yl- electron donor fragments. 2-(Aryl/thiophen-2-yl)quinazolin-4(3H)-ones showed fluorescence in blue-green region in solutions, triphenylamino derivative with para-phenylene linker displayed the highest quantum yield of 89% in toluene solution and 39.7% in powder. Strong influence of donor fragment nature on photophysical properties was observed, the fluorescence intensity of Et_2_N and Ph_2_N derivatives decreased when going from toluene to MeCN solution, whereas carbazol-9-yl counterparts demonstrated growing of QY. 4-Cyanoquinazolines are less emissive both in solutions and solid state than their quinazolin-4-one counterparts. The introduction of cyano group led to orange/red colored powder and dual emission bands. Emission intensity of some molecules was enhanced upon the addition of water to MeCN solution. 4-Cyanoquinazolines exhibited enhanced intramolecular charge transfer regarding their quinazolin-4-one analogues as was shown by their red-shifted absorption and emission spectra as well as their decreased electrochemical gap by more than 1 eV.

## Data Availability

The data are available on request from the corresponding authors.

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
