# Peer review of "Push-Pull Structures Based on 2-Aryl/thienyl Substituted Quinazolin-4(3H)-ones and 4-Cyanoquinazolines"

_molecules, 2022, doi:10.3390/molecules27217156_

Round 1
Reviewer 1 Report
The authors are reporting a comparative study on the photochemical properties of 2-substituted quinazolinones and 4-cyanoquinazolines. While being a routine study on the comparison of 17 molecules (13 of them being new), the study was conducted on a systematic manner and the synthesis of the new entities properly described.
The discussion of the results is coherent and logical, apart from some typos that can be found along the text.
1H NMR spectra for compounds 8b and 9, free from solvent traces, should be added.
Author Response
1) We made some corrections:
Line 97 – arylboronic acid
Line 124 – angles
Line 129 – demonstrates
Line 146 – their counterparts
Line 185 – leads
Line 215 – luminescent
Line 228 – 1,4-phenylene
Line 299 and 308 – gradual
Line 304 – possesses
Line 347 – compounds
Line 388 - cubic angstrom
Moreover, we corrected prepositions, punctuation and made some other improvements.
2) to comment "1H NMR spectra for compounds 8b and 9, free from solvent traces, should be added".
The 1H NMR spectra for compounds 8b and 9 were registered after purification by recrystallization from СН2Cl2/hexane and indeed they contained the signals of the solvents. The signals of solvents disappeared after carefully drying of samples under air. We changed the corresponding spectra in SI.
Reviewer 2 Report
The manuscript written by Dr. Nosova and the team explains synthesis of various quinazoline derivatives and their photophysical properties. The manuscript is well-written. It is highly suitable for publication in journal ‘’molecules’’. Some minor corrections are required before final acceptance (minor revision).
- ‘’of’’ should be replaced with ‘’the’’ (2nd page, 2nd paragraph, 2nd line) (…to investigate the structure-property relations..)
- ‘’is’’ should be replaced with ‘’in’’ (2nd page, 2nd paragraph, 5th line) (We succeeded in deepening..)
- ‘’then’’ should be replaced with ‘’than’’ (Conclusion, 10th line)
- The authors should carefully read the grammatical errors throughout the text.
Author Response
- ‘’of’’ should be replaced with ‘’the’’ (2nd page, 2nd paragraph, 2nd line) (…to investigate the structure-property relations.)
We replaced ’of’’ with ‘’the’’
- ‘’is’’ should be replaced with ‘’in’’ (2nd page, 2nd paragraph, 5th line) (We succeeded in deepening.)
We replaced ’is’’ with ‘’in’’
- ‘’then’’ should be replaced with ‘’than’’ (Conclusion, 10th line)
We replaced ‘’then’’ with ‘’than’’
- The authors should carefully read the grammatical errors throughout the text.
We made some corrections:
Line 97 – arylboronic acid
Line 124 – angles
Line 129 – demonstrates
Line 146 – their counterparts
Line 185 – leads
Line 215 – luminescent
Line 228 – 1,4-phenylene
Line 299 and 308 – gradual
Line 304 – possesses
Line 347 – compounds
Line 388 - cubic angstrom
Moreover, we corrected prepositions, punctuation and made some other improvements.
Reviewer 3 Report
In this work, Nosova et al. reported the synthesis and photophysical studies of a series of 2-(aryl/thiophen-2-yl)quinazolin-4(3H)-ones and 4-cyano-2-arylquinazolines. The compounds are well characterized by NMR, MS, and EA, and the purity observed across the NMR data provided is excellent. This is a nice and comprehensive photophysical study, I agree with most of the assertions and conclusions. Thus, it would be an interesting addition to the related area and warrants publication in Molecules after addressing some of the questions and comments outlined below:
1. Line 108, the author stated that’ due to its poor stability caused by the strong electron-donating effect of triphenylene residue’, it is kind of confusing. Is that what the author was trying to convey that the strong electron-donating effect of the TPA unit causes the chlorination step to be sluggish, leading to the low yield of 11?
2. Figure 3b is pretty dark, should be fixed to be a better view.
3. The photophysical properties were only studied in two solvents, it would be good to study solvatochromism for these ICT fluorophores.
4. For dual-emission fluorophores, the author mentioned that the double peaks originated from the possible excimer or aggregation of molecules. The reviewer suggests that the dilution experiment could be helpful in this case.
5. For Figure 5b, the caption ‘UV/Vis (dash lines) and photoluminescence (solid lines)’ seems to be wrong, the UV/vis are actually shown in the dashed line.
6. Line 338, ‘The HOMO-LUMO levels of 4-cyanoquinazolins are practically not overlapped in contrast to quinazolinones energy levels’ is misleading.
If I was right, this could go like this: When compared to the quinazolinones derivatives, the HOMO and LUMO distributions of 4-cyanoquinazolins are barely overlapped.
7. In our lab, we use the same instrument model FS5 and the same software Fluoracle. The IRF obtained in the lifetime measurements is somewhat wired in most cases. Did the author use the Ludox when measuring the IRF?
*ESI
In the Quantum-chemical calculations part,
Table S4, E0-0 should be the Egap calculated from DFT, 0-0 energy is a different concept, which denotes the energies for the electronic transitions from the lowest vibrational states of S1 and S0.
Author Response
1. Line 108, the author stated that’ due to its poor stability caused by the strong electron-donating effect of triphenylene residue’, it is kind of confusing. Is that what the author was trying to convey that the strong electron-donating effect of the TPA unit causes the chlorination step to be sluggish, leading to the low yield of 11?
We corrected this point as follows:
"We failed to isolate the intermediate 4-chloroquinazoline. Moreover, the yield of target product 11 was 28%. Probably the strong electron-donating effect of the TPA unit causes the chlorination step to be sluggish, leading to the low yield of 11".
2. Figure 3b is pretty dark, should be fixed to be a better view.
We replaced the Figure 3b into another one for a better viewing.
3. The photophysical properties were only studied in two solvents, it would be good to study solvatochromism for these ICT fluorophores.
We chose two solvents with different polarity to compare with quinazoline derivatives previously studied by our group. We are planning deeper study (including solvatochromism, lifetime calculations and so on) of more outstanding samples. This work is under progress.
4. For dual-emission fluorophores, the author mentioned that the double peaks originated from the possible excimer or aggregation of molecules. The reviewer suggests that the dilution experiment could be helpful in this case.
As in the case of previously mentioned point, we are working on the detailed study of the fluorescence mechanisms of quinazoline compounds, paying attention to the observation of two emission peaks. We are going to prepare the separate manuscript on 4-morpholinylquinazolines, 4-oxoquinazolines and 4-cyanoquinazolines next year to report an advanced research.
Nevertheless, we can note concerning the current manuscript, that in the case of quantum yield and lifetime measurements the dilution was very strong (absorption up to 0.1). We observed the same peaks and, no situations when the peak disappeared or shifted were revealed, the intensity was consistent with the concentration.
5. For Figure 5b, the caption ‘UV/Vis (dash lines) and photoluminescence (solid lines)’ seems to be wrong, the UV/vis are actually shown in the dashed line.
Indeed, we corrected the caption.
6. Line 338, ‘The HOMO-LUMO levels of 4-cyanoquinazolins are practically not overlapped in contrast to quinazolinones energy levels’ is misleading.
If I was right, this could go like this: When compared to the quinazolinones derivatives, the HOMO and LUMO distributions of 4-cyanoquinazolins are barely overlapped.
Thank you for the correction. We changed the statement into “When compared to the quinazolinones derivatives, the HOMO and LUMO distributions of 4-cyanoquinazolins are barely overlapped.”
+ 7. In our lab, we use the same instrument model FS5 and the same software Fluoracle. The IRF obtained in the lifetime measurements is somewhat wired in most cases. Did the author use the Ludox when measuring the IRF?
No, we did not use Ludox when measuring the IRF. We use quartz frosted glass or water. Both are recommended directly by the FS5 manufacturer and are listed in their guidelines.
- ESI In the Quantum-chemical calculations part,
Table S4, E0-0 should be the Egap calculated from DFT, 0-0 energy is a different concept, which denotes the energies for the electronic transitions from the lowest vibrational states of S1 and S0.
We corrected the caption of column into “Energy band gap (ΔE)”.